# Production of Microsclerotia by *Metarhizium* sp., and Factors Affecting Their Survival, Germination, and Conidial Yield

**DOI:** 10.3390/jof8040402

**Published:** 2022-04-14

**Authors:** Meelad Yousef-Yousef, Antonia Romero-Conde, Enrique Quesada-Moraga, Inmaculada Garrido-Jurado

**Affiliations:** Departamento de Agronomía (DAUCO María de Maeztu Unit of Excellence 2021–2023), Campus de Rabanales, Universidad de Córdoba, Edif. C4, 14071 Córdoba, Spain; g92rocoa@uco.es (A.R.-C.); cr2qumoe@uco.es (E.Q.-M.); g72gajui@uco.es (I.G.-J.)

**Keywords:** entomopathogenic fungi, temperature, moisture, UV-B, soil texture, resistance structures

## Abstract

Microsclerotia (MS) produced by some species of *Metarhizium* can be used as active ingredients in mycoinsecticides for the control of soil-dwelling stages of geophilic pests. In this study, the MS production potential of two *Metarhizium brunneum* strains and one *M. robertsii* strain was evaluated. The three strains were able to produce MS in liquid fermentation, yielding between 4.0 × 10^6^ (*M. robertsii* EAMa 01/158-Su strain) and 1.0 × 10^7^ (*M. brunneum* EAMa 01/58-Su strain) infective propagules (CFU) per gram of MS. The EAMa 01/58-Su strain was selected for further investigation into the effects of key abiotic factors on their survival and conidial yield. The MS were demonstrated to be stable at different storage temperatures (−80, −18, and 4 °C), with a shelf-life up to one year. The best temperature for MS storage was −80 °C, ensuring good viability of MS for up to one year (4.9 × 10^10^ CFU/g MS). Moreover, soil texture significantly affected CFU production by MS; sandy soils were the best driver of infective propagule production. Finally, the best combination of soil temperature and humidity for MS germination was 22.7 °C and 7.3% (*wt.*/*wt.*), with no significant effect of UV-B exposure time on MS viability. These results provide key insights into the handling and storage of MS, and for decision making on MS dosage and timing of application.

## 1. Introduction

Mycoinsecticide development technologies for the entomopathogenic genus *Metarhizium* have been based largely on solid-state fermentation and the production of aerial conidia. However, the mass production of viable conidia faces several challenges, namely, improving temperature, oxygen, pH, and humidity conditions during mass production and harvesting; reducing solid waste; and reducing the labor requirements and production time [1,2,3,4].

Microsclerotia (MS) are desiccation-tolerant survival structures in ascomycetes and basidiomycetes. They are compact, melanic, solid hyphae aggregated into particles, with a size range of 200–600 µm in the genus *Metarhizium* [5]. MS are capable of surviving in latency under adverse conditions [6]. MS may provide an alternative to *Metarhizium* spp. conidia for mycoinsecticide development against geophilic insects of agricultural, medical, and veterinary importance [7,8,9,10,11]. Indeed, MS can be obtained via liquid fermentation, without the drawbacks of solid fermentation, and they can produce infective conidia after rehydration [7,8,9,10,12,13].

MS storage conditions may influence germination potential and conidial yield, which are also affected by fungal species, fungal strain, formulation, and packaging [14,15,16,17,18,19,20]. Indeed, soil type and soil abiotic and biotic conditions may strongly influence the viability and availability of entomopathogenic fungal propagules [21,22,23]. With respect to conidia, previous research has shown that soils with lower clay content and higher sand content are better for the survival of *Metarhizium* conidia [21]. Temperature and humidity are the main factors governing entomopathogenic fungi’s viability and virulence [24,25].

The use of MS for mycoinsecticide development is an innovative strategy for the release of resistant structures capable of multiplying, producing, and releasing large quantities of infective propagules in the target environment and, even under unfavorable conditions in soil, a bank of conidia for future insect outbreaks [7,9,26,27]. However, MS handling and storage conditions before application in the field require investigation. Furthermore, the development of decision-support tools that optimize MS dosage and application timing requires a better understanding of the effects of storage conditions, incubation time, and soil texture, temperature, and humidity conditions on survival, germination, and conidial yield. In the present study, the potential of two *Metarhizium brunneum* strains and one *M. robertsii* strain to produce MS in liquid media was evaluated, and the effects of MS storage conditions, incubation time, and soil conditions on their survival were investigated.

## 2. Materials and Methods

### 2.1. Fungal Species and Strains Used in This Study

Two *M. brunneum* strains (EAMb 09/01-Su and EAMa 01/58-Su) and one *M. robertsii* strain (EAMa 01/158-Su) were obtained from the culture collection of the Agricultural Entomology Research Group AGR 163, from the Department of Agronomy of the University of Cordoba. All of them were selected from different habitats and locations (Table 1) and have been described previously as being pathogenic against soil-dwelling insects [21,28,29,30,31,32].

### 2.2. Production of Microsclerotia by M. brunneum and M. robertsii Strains, and Their Germination and Infective Propagule Yield

The liquid medium used for MS production was composed as follows (all values were per liter of deionized water): 75.0 g of glucose (Carbo Erba Reagents, Barcelona, Spain), 15.0 g of acid-hydrolyzed casein (PhytoTechnology Laboratories, Shawnee Mission, KS, USA), 2.0 g of KH_2_PO_4_ (Panreac, Barcelona, Spain), 0.4 g of CaCl_2_.2H_2_O (Panreac, Barcelona, Spain), 0.3 g of MgSO_4_.7H_2_0 (Duchefa Biochemie, Haarlem, The Netherlands), 0.05 g of FeSO_4_.7H_2_O (Panreac, Barcelona, Spain), 0.037 g of CoCl_2_.6H_2_O (Acros Organics^TM^, Morris Plains, NJ, USA), 0.016 g of MnSO_4_.H_2_O (Fisher Scientific, Loughborough, UK), 0.014 g of ZnSO_4_.7H_2_O (VWR International Eurolab, Barcelona, Spain), 5.0 × 10^−4^ g of thiamine (VWR International Eurolab, Barcelona, Spain), 5.0 × 10^−4^ g of riboflavin (VWR International Eurolab, Barcelona, Spain), 5.0 × 10^−4^ g of pantothenate (Alfa Aesar, Kandel, Germany), 5.0 × 10^−4^ g of niacin (Merck, Darmstadt, Germany), 5.0 × 10^−4^ g of pyridoxamine (Biosynth Carbosynth^®^, Compton, Berkshire, UK), thioctic acid (Merck, Darmstadt, Germany), 5.0 × 10^−5^ g of folic acid (Merck, Darmstadt, Germany), 5.0 × 10^−5^ g of biotin (Biosynth Carbosynth^®^, Compton, Berkshire, UK), and 5.0 × 10^−5^ g of vitamin B_12_ (Alfa Aesar, Kandel, Germany) [33,34]. Six flasks per strain were inoculated with a 1.0 × 10^7^ conidia/mL suspension and incubated at 28 °C and 300 rpm in a rotary shaker incubator for MS production. The flasks were then fermented for 10 days.

Evaluation of MS production was carried out as follows: First, 100 µL of the fermented liquid medium was observed under an optical microscope (Motic BA400, Barcelona, Spain) (Figure 1). To determine MS germination rates, 100 µL of fermented liquid medium was incubated in 9 cm Petri plates (APTACA S.p.A., Asti, Italy) containing a water–agar medium (2% *wt.*/*v*) (*n* = 6 per strain) and incubated for 10 days at 28 °C, after which the proportion of MS germinating was determined [13,27].

After that, diatomaceous earth (Scharlab S.L., Barcelona, Spain) was added to the fermented liquid medium at a rate of 5% (*wt.*/*v*), and the cultures were filtered under vacuum through Whatman N° 54 filter paper in a Buchner funnel. The solid fraction after filtration was broken up by pulsing in a blender (Mini Prep^®^ Plus, Cuisinart), layered in shallow aluminum trays, and air-dried overnight in the airflow within a biological containment hood, to a final moisture content of ≤5% (gravimetric). The dry material was passed through an N° 120-mesh sieve to produce granular particles of 0.6–1.5 mm in diameter. These MS-containing granules were stored in tubes at 5–7 °C. Then, 0.1 g of the solid product obtained was sprinkled onto malt agar (MA) (Biolife, Milano, Italy) plates (*n* = 3 per strain), incubated for 10 days at 28 °C, and the colony-forming units (CFU) per gram of MS were determined for each strain.

The number of CFU per g was calculated using the following formula:CFU/mL = Number of CFU counted × dilution × 10
Number of CFU/g MS = CFU/mL × 10 (mL) × 10

### 2.3. Effect of Storage Temperature on Microsclerotia Germination and Infective Propagule Yield of M. brunneum Strain EAMa 01/58-Su

The effects of storage temperature on the shelf-life of MS and their capacity to germinate and produce conidia were evaluated. For this, the MS were stored at 25 °C, 4 °C, −18 °C, and −80 °C, and evaluated every 4 months for 12 months. For each storage temperature and evaluation time, 0.1 g of the stored MS was incubated on 9 cm Petri plates in a water–agar medium (2% *wt./v*) at 25 °C for 7 days in darkness. After this time, the viability of the conidia produced by the MS was determined (*n* = 3 per temperature). To determine the number of conidia/g of MS, germinated MS (with conidia) were scraped from Petri plates into a sterile solution of 0.1% Tween 80, sonicated (Ultrasons HD 3,000,865, J.P. Selecta S.A., Barcelona, Spain) for 5 min, and then filtered through several layers of sterile gauze. Then, serial dilutions were performed. Conidia concentration was determined with a hemocytometer (Malassez chamber, Blau Brand, Wertheim, Germany).

### 2.4. Effect of Incubation Time on Microsclerotia Germination and Infective Propagule Yield of M. brunneum Strain EAMa 01/58-Su

To determine the initiation of infective propagule production by MS of the *M. brunneum* EAMa 01/58-Su strain, along with the effect of incubation time on the production of infective propagules, seven consecutive 48-h time intervals (24, 72, 120, 168, 216, 264, and 336 h) were evaluated. For that, 0.1 g of MS was incubated in 9 cm Petri plates of a selective culture medium composed of 65.0 g of SDA (EMD Chemicals Inc., Gibbstown, NJ, USA), 0.5 g of amphenicol chlorine (Sigma-Aldrich, Steinheim, Germany), 0.25 g of cycloheximide (Sigma-Aldrich, Steinheim, Germany), and 0.01 g of dodine (Chemservice, West Chester, PA, USA) per liter of water, and incubated for 10 days at 25 °C (*n* = 3 per time interval).

At each incubation time, the conidia from each plate were scraped off and diluted with 10 mL of 0.1% aqueous Tween 80, and then separated by ultrasound. To determine the number of CFU, 100 μL of each serial dilution of conidia from germinated MS was spread onto malt agar (MA) medium, sealed with Parafilm, and incubated for 5–7 days at 25 °C (*n* = 3 per dilution). The numbers of CFU were counted in three plates per dilution and soil type, as described previously (Section 2.2).

### 2.5. Effect of Soil Texture on Microsclerotia Germination and Infective Propagule Yield of M. brunneum Strain EAMa 01/58-Su

To study the effect of soil texture on the production of infective propagules by MS of the *M. brunneum* EAMa 01/58-Su strain, five different soils from the collection at the Department of Agronomy of the University of Córdoba (Spain) were compared. The properties of these soils are shown in Table 2.

All of the soils were maintained at field capacity by adding sterile water and 10 g of each sterilized soil to 5 cm Petri plates. Then, 0.1 g of *M. brunneum* EAMa 01/58-Su MS was added to each plate. The evaluation of infective propagule production and its viability was carried out weekly for 12 weeks (*n* = 3 per soil type and week). Plates were sealed and incubated at 25 °C in darkness. Every week three plates per soil were taken from the incubator, and the soil of these plates was mixed with 20 mL of 0.1% aqueous Tween 80. The mixture was vigorously stirred to separate conidia from the soil particles, and then serial dilutions were prepared. The numbers of CFU were counted (*n* = 3 per 1/100 dilution and soil type) as described previously (Section 2.2).

### 2.6. Effects of Soil Temperature and Moisture on Microsclerotia Germination and Infective Propagule Yield of M. brunneum Strain EAMa 01/58

The soil used in the experiment was collected from the field in the province of Córdoba (Spain), and was characterized as sandy clay (78.0% sand, 17.0% silt, 5.0% clay, and 0.2% organic matter) with a pH of 8.3. The soil was sieved through a 2 mm light sieve and stored dry at 25 °C. It was then sterilized at 121 °C for 20 min and dried in an oven at 105 °C for 24 h. Soil (30.0 g) was added to each Petri plate, and then different amounts of sterile water were added to achieve different degrees of moisture in the soil. The amounts of water added per plate were 0.3, 1.5, 2.7, 3.9, and 5.1 mL to achieve the following moistures: −2.14 MPa [1% (*wt.*/*wt.*)], −0.5 MPa [5.0% (*wt.*/*wt.*)], −0.47 MPa [9.0% (*wt.*/*wt.*)], −0.28 MPa [13.0% (*wt.*/*wt.*)], and −0.23 MPa [17.0% (*wt.*/*wt.*)], respectively [25]. Then, 0.1 g of MS was added to each plate and incubated in darkness at 15, 20, 25, 30 or 35 °C) to achieve different combinations of temperature and moisture. In total, five Petri plates were used for each combination of temperature and moisture. After 8 days of incubation, germination and CFU production were evaluated. Germination was evaluated by examining 10 MS from each plate with a magnifying glass to verify whether they were germinated or not. MS were considered to have germinated when mycelium appeared radially on them, continuing to develop until conidiogenesis began. Evaluation of infective propagule yield was performed for all temperature–moisture combinations after 8 days; the contents of each Petri plate were emptied into a 15.0 mL sterilized tube, and 20.0 mL of 0.1% aqueous Tween 80 was added. The contents of each tube were vigorously shaken to separate the conidia from the MS and soil particles. Serial dilutions were carried out as described previously to facilitate the counting. Then, 0.1 mL of each dilution was spread on Petri plates with selective medium and incubated at 25 °C in darkness for 8 days. The numbers of CFU were calculated using the formula described in Section 2.2.

### 2.7. Ultraviolet Radiation (UV-B) Effects on Microsclerotia Germination and Infective Propagule Yield of M. brunneum Strain EAMa 01/58-Su

The effect of UV-B radiation on the germination of MS and subsequent yield of infective propagules was studied using five exposure times: 4, 8, 24, 48, and 72 h. To each 9 cm Petri plate, 30.0 g of sandy clay soil (78.0% sand, 17.0% silt, 5.0% clay, and 0.2% organic matter) was added. Then, 2.2 mL (7.3% *wt.*/*wt.*) of sterile water was added to the soil. Finally, 0.1 g of MS was added to the soil surface in each Petri plate. There were three replicate Petri plates for each UV-B exposure time and control. The Petri plates were exposed to radiation of 1200 mWm^−2^ in a temperature-controlled chamber (Fitoclima S600PL, ARALAB, Portugal). The temperature (23 ± 1 °C) and moisture (7.3% *wt.*/*wt.*) conditions used were selected according to the results obtained in the previous experiment (Section 2.6).

The irradiated samples were protected by a 0.13 mm thick cellulose diacetate film, which allowed entry of UV-B and UV-A radiation (λ > 315 nm), but not UV-C (λ < 280 nm); the control treatment was wrapped in aluminum foil to exclude radiation [35].

After each radiation exposure time, the Petri plates were incubated at 25 °C in darkness for 8 days. Afterwards, CFU production and germination were evaluated as described previously (Section 2.2).

### 2.8. Statistical Analysis

Statistical analyses were performed using Statistix 9^®^ (Analytical Software, Tallahassee, FL, USA). Analysis of variance (ANOVA) was used to compare differences in MS production between strains of *Metarhizium*, as well as differences in infective propagule production by MS in relation to soil texture and UV-B exposure. When significant treatment effects were identified, means were compared using Tukey’s HSD test. The effects of storage time and temperature on infective propagule production by the MS were analyzed using a linear mixed model for repeated measures (split plot in time):
*Log*10 (*conidia +* 1) = *Temperature* + *Time* (*quarter*) + *Temperature* × *Time* (*quarter*) + *Temperature* × *Repetition*

The number of conidia was *log*10 (*x* + 1) transformed to meet the normality and homogeneity of the variance assumptions. Temperature, time (quarter), and their interaction were modelled as fixed effects, and temperature × repetition was modelled as a random effect. The model was estimated using the restricted maximum likelihood (REML) method, and means were compared using Tukey’s test (α = 0.05) [36]. Mean ± standard error was used to detect relationships between the production of conidia by the MS of the *M. brunneum* EAMa 01/58-Su strain and incubation time.

The effects of soil temperature and moisture on the numbers of CFU produced by *M. brunneum* MS were evaluated using a generalized linear model with logarithmic function and Poisson distribution:
*Log* (*CFU* 1:10000) = *β*_0_ + *β*_1_*Temperature* + *β*_2_*Moisture* + *β*_3_*Temperature* × *Moisture* + *β*_4_*Temperature*^2^ + *β*_5_*Moisture*^2^

The model was estimated using the maximum likelihood method. The interaction term (*Temperature* × *Moisture*) was not statistically significant and was removed from the model.

## 3. Results

### 3.1. Production of Microsclerotia, Germination, and Infective Propagule Yield by M. brunneum and M. robertsii Strains

There were no significant differences between strains in MS production per liter of liquid medium (F_2,11_ = 0.56; *p* = 0.5879), which ranged between 4.6 × 10^11^ and 8.2 × 10^11^ MS/L for the EAMa 01/58-Su and EAMa 01/158-Su strains, respectively. However, significant differences were found between strains in the numbers of CFU produced per gram of MS, with the EAMa 01/58-Su strain showing the highest number at 1.0 × 10^7^ CFU/g MS (F_2,8_ = 14.35; *p* = 0.015) (Figure 2).

### 3.2. Effects of Storage Temperature on Microsclerotia Germination and Infective Propagule Yield of M. brunneum Strain EAMa 01/58-Su

Temperature and storage time significantly affected MS shelf-life (F_3,8.4_ = 6.07; *p* = 0.02 and F_2,15.1_ = 59.10; *p* < 0.0001 respectively). However, MS shelf-life was not significantly affected by the interaction between temperature and time (F_6,15.04_ = 2.66; *p* = 0.06). Throughout the 12-month study period, significant differences in conidia production were found only at the first 4-month incubation period, with a higher yield at −80 °C (4.9 × 10^10^ conidia/g MS in quarter 1) than at the other temperatures (Figure 3).

### 3.3. Effects of Incubation Time on Microsclerotia Germination and Colony Yield of M. brunneum Strain EAMa 01/58-Su

Throughout the selected range of incubation periods, CFU production started at 168 h after MS incubation, with a quantity of 4.0 × 10^8^ CFU/g MS. Maximum production was achieved at 264 h, with 7.3 × 10^9^ CFU/g MS. From this day onwards, production of CFU progressively decreased (Figure 4).

### 3.4. Effects of Soil Texture on Microsclerotia Germination and Colony Yield of M. brunneum Strain EAMa 01/58-Su

The effect of soil texture on CFU production from MS of the *M. brunneum* strain EAMa 01/58-Su was significant (F_4,175_ = 3.58; *p* < 0.01). Sandy soil textures (AG35 and AG39) were the best drivers of CFU production by this strain, with higher values for CFU detected during the first week compared to other soils (AG4, AG6, and AG20) (F_4,10_ = 115.28; *p* < 0.001) (Figure 5). For all soils, the highest CFU yield was detected between the 8th and 9th weeks, ranging between 1.4 × 10^9^ CFU/g of MS for AG4 soil and 5.3 × 10^9^ CFU/g of MS for AG39.

### 3.5. Effects of Temperature and Moisture on Microsclerotia Germination and Colony Yield of M. brunneum Strain EAMa 01/58-Su

The generalized linear model used to evaluate the effects of temperature and soil moisture on the numbers of CFU produced by the MS of strain EAMa 01/-Su was statistically significant ($\chi^{2}$_(4)_ = 85.47, *p* < 0.0001). Linear and quadratic terms for both temperature and soil moisture were statistically significant (Table 3). The optimal combination of temperature and moisture for maximal production of CFU per gram of MS was at a temperature of 22.7 °C and a soil moisture of 7.28% (*wt.*/*wt.*) (Figure 6).

The temperature × moisture interaction was not significant and was excluded from the model.

### 3.6. Effects of Ultraviolet Radiation (UV-B) on Microsclerotia Germination and CFU Yield of M. brunneum Strain EAMa 01/58-Su

There was no significant effect of MS exposure to UV-B 1200 mWm^−2^ on their germination and CFU yield (F_5,28_ = 1.61; *p* = 0.188), which ranged between 6.0 × 10^7^ and 9.1 × 10^8^ CFU/g MS (Table 4).

## 4. Discussion

Entomopathogenic fungi (EPF) are soil-inhabiting microorganisms widely used against soil-dwelling stages of insect pests for sustainable crop production [32,37]. EPF application for pest control has been based mainly on the use of conidia, while using MS has several advantages for the development of cost-competitive microbial control formulations. Given the capacity of MS to persist in soils and decaying plant material under unfavorable conditions, these propagules often serve as a reservoir of infective conidia that can become active under favorable conditions [11,27]. MS applied to the soil as a granular formulation yield infective conidia after rehydration, with abiotic soil conditions being key factors in their effective use [38]. In this study, we evaluated the ability of three strains of *Metarhizium* (*M. brunneum* strains EAMb 09/01-Su and EAMa 01/58-Su, and *M. robertsii* strain EAMa 01/158-Su) to produce MS. All strains are well known for their efficacy when applied as conidia against soil-dwelling insect pests [11,12,13,14,15,16,17,18,19,20,21,22,23,24,25,26,27,28], and even against plant diseases [9]. We also evaluated the effects of abiotic factors and soil conditions on shelf-life/persistence and the capacity of the MS to produce infective conidia.

The three *Metarhizium* strains all produced high yields of MS in liquid fermentation (up to 8.2 × 10^11^ MS/L) [39], and achieved high values for CFU production per gram of MS; the EAMa 01/58-Su strain achieved the highest rate, with 1.0 × 10^7^ CFU/g MS. This rate is twice as high as the recently recorded MS production by other *Metarhizium* spp. fermented under the same conditions [10,40]. As observed in our study, a high yield of MS/L of fermentation medium was recorded 10 days after the initiation of liquid fermentation, as demonstrated elsewhere [10], which may indicate similar tendencies across *Metarhizium* species in this trait.

The strain EAMa 01/58-Su produced more CFU/g of MS than the other strains (1.0 × 10^7^ CFU/g MS) used in this study. Production of infective propagules could also be increased by replacing the diatomaceous earth carrier with vermiculite, as previously reported [41]. The capacity of some EPF strains to produce MS is considered a key factor for future use at a commercial level. However, EPF MS must be able to yield high numbers of conidia after rehydration, since conidia are necessary to initiate infection [22]. In fact, these results indicate the potential of the strain EAMa 01/58-Su to produce MS, as well as the capacity of these MS to provide large numbers of infective conidia, which are known to have the potential to control fruit fly (Diptera; Tephritidae) preimaginals in the soil [25,32].

The maximum yield of conidia by the strain EAMa 01/58-Su MS (7.3 × 10^9^ CFU/g) occurred at 11 days after incubation, as previously shown by other researchers [11,42]. This suggests that if these MS are to be used as a possible alternative to conidia in soil treatments against *B. oleae* preimaginals in the soil [30,31,32,37], they must be applied to the soil at least 11 days earlier than conidia would be applied, so as to ensure that the required numbers of conidia are present when third-instar *B. oleae* larvae drop to the soil.

However, our results also showed high stability of the strain EAMa 01/58-Su MS at different storage temperatures, with significant production of infective propagules after storage of up to one year, which confirms earlier findings [43]. This viability at relatively high storage temperatures (4 °C and 25 °C) provides post-production opportunities for companies and farmers to store the product prior to sale or application. In general, this strain shows similar trends in terms of storage stability as other *Metarhizium* strains [13,18]. Despite this, the present work also showed that the most suitable temperature for MS germination and infective conidia production was 22.7 °C, with a soil moisture content of 7.28% (*wt.*/*wt.*). This optimal combination of temperature and moisture for MS germination is slightly different than the optimal temperature for conidiogenesis, which was previously set at a temperature of 26.3 °C [44]. The effects of temperature and moisture on the germination and growth of EPF propagules is a key point to be evaluated on a case-by-case basis, due to intraspecific differences that may exist between strains of the same fungal species. This is the case for *Metarhizium*, where the range of temperatures established for growth vary between 22 °C and 30 °C [45,46,47,48], while the best temperature for conidial germination varies between 15 °C and 35 °C [49]. This needs to be considered when developing MS for pest control; data indicate that after MS application a slight modification of the environment (e.g., regulated irrigation to increase humidity) can enhance infective conidia yield and, therefore, the success of the control method. The temperature–moisture combinations needed for MS germination and high infective conidia yield suggest that our control strategy for tephritids based on autumn and spring applications of conidia of the same strain is also valid for MS applications, because climatic conditions are suitable for soil applications of both MS and conidia during these seasons [32,37].

For all of the soil textures evaluated, the maximum colony yield by the MS occurred at the 8th week, which was a longer period than reported for *Metarhizium* conidia germination in soils of different textures [50]. The effects of soil texture on MS germination and infective conidia production were the main drivers of CFU production, with higher values of infective propagules produced in sandy soils than in clay soils. The effect of soil texture on fungus survival—especially conidia—is well known, and many studies have shown that high sand content is associated with greater fungal survival compared with clay soil [50]. Other studies have also shown that the viability of *Metarhizium* conidia is higher in sandy soils than in clay soils [21,51]. In general, soil texture and structure are important factors that affect the availability of nutrients, oxygen, water, and shelter for EPF [38,52]. Moreover, high sand content facilitates the movement of water and the diffusion of air, as well as the movement of fungal propagules [53]. Finally, the time of exposure to UV-B radiation had no effect on the capacity of the *M. brunneum* strain EAMa 01/58-Su MS to produce infective conidia, showing a high tolerance to UV-B, as previously described [54]. Previously reported oxidative stress during MS formation could be involved in the observed UV-B tolerance [55]. Degradation of conidia by UV-B radiation is likely to reduce insect mortality [35,37]; in contrast, the higher environmental resilience of MS makes them more likely to survive in the soil until conditions that are favorable for insect infestation. Soil is the natural habitat of EPF, providing them with protection from environmental extremes, and is therefore generally considered to be a good environment for long-term fungal microbial control strategies for geophilous insects.

These results shed light on the MS produced by *Metarhizium* sp. and provide key insights for the handling and storage of MS, as well as for decision making with regard to MS dosage and timing of application.

## Figures and Tables

**Figure 1 jof-08-00402-f001:**
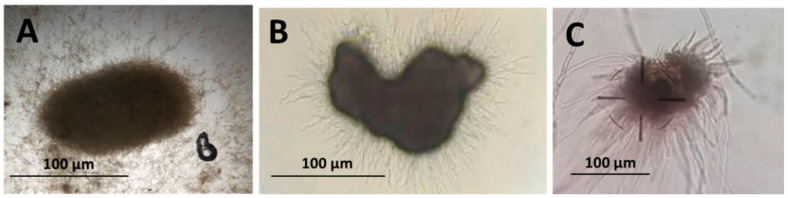
Microsclerotia of *Metarhizium* sp. observed after 10 days in liquid fermentation at 28 °C and 300 rpm: (**A**) microsclerotium of *M. robertsii* EAMa 01/158-Su strain; (**B**) microsclerotium of *M. brunneum* EAMb 09/01-Su strain; (**C**) microsclerotium of *M. brunneum* EAMa 01/58-Su strain. The photomicrograph was taken at 100× magnification, using an optical microscope (Motic BA400, Barcelona, Spain) and a digital camera (Leica DFC 450, Barcelona, Spain).

**Figure 2 jof-08-00402-f002:**
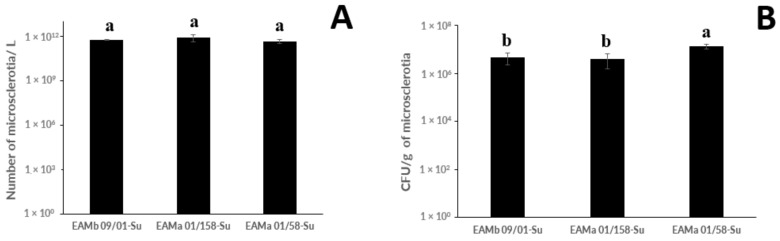
(**A**) Microsclerotia production in liquid medium after 8 days at 25 °C; (**B**) CFU produced per gram of microsclerotia. Mean values (±SE) followed by different letters are significantly different to one another according to Fisher’s protected Tukey’s HSD test (*p* ≤ 0.05).

**Figure 3 jof-08-00402-f003:**
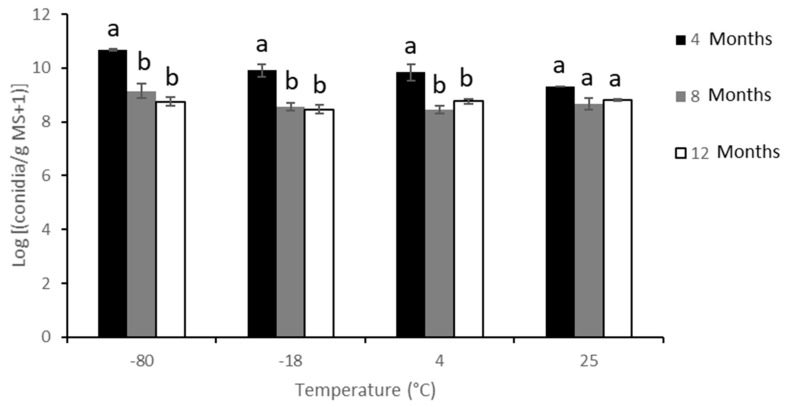
Conidia yield per gram of *M. brunneum* EAMa 01/58-Su strain microsclerotia after 4, 8, and 12 months at −80 °C, −18 °C, 4 °C, and 25 °C. Mean values (±SE) within temperature followed by different letters are significantly different to one another according to the mean separation test with 95% protected by Tukey’s HSD range test (*p* ≤ 0.05).

**Figure 4 jof-08-00402-f004:**
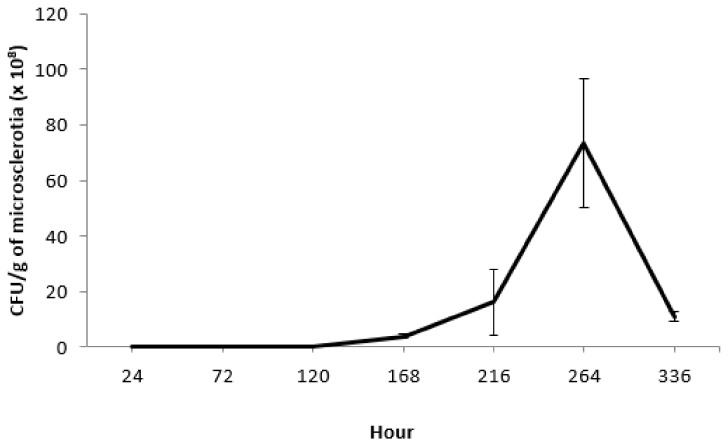
Time course of CFU yield (mean ± SE) of *M. brunneum* strain EAMa 01/58-Su microsclerotia as a function of incubation time.

**Figure 5 jof-08-00402-f005:**
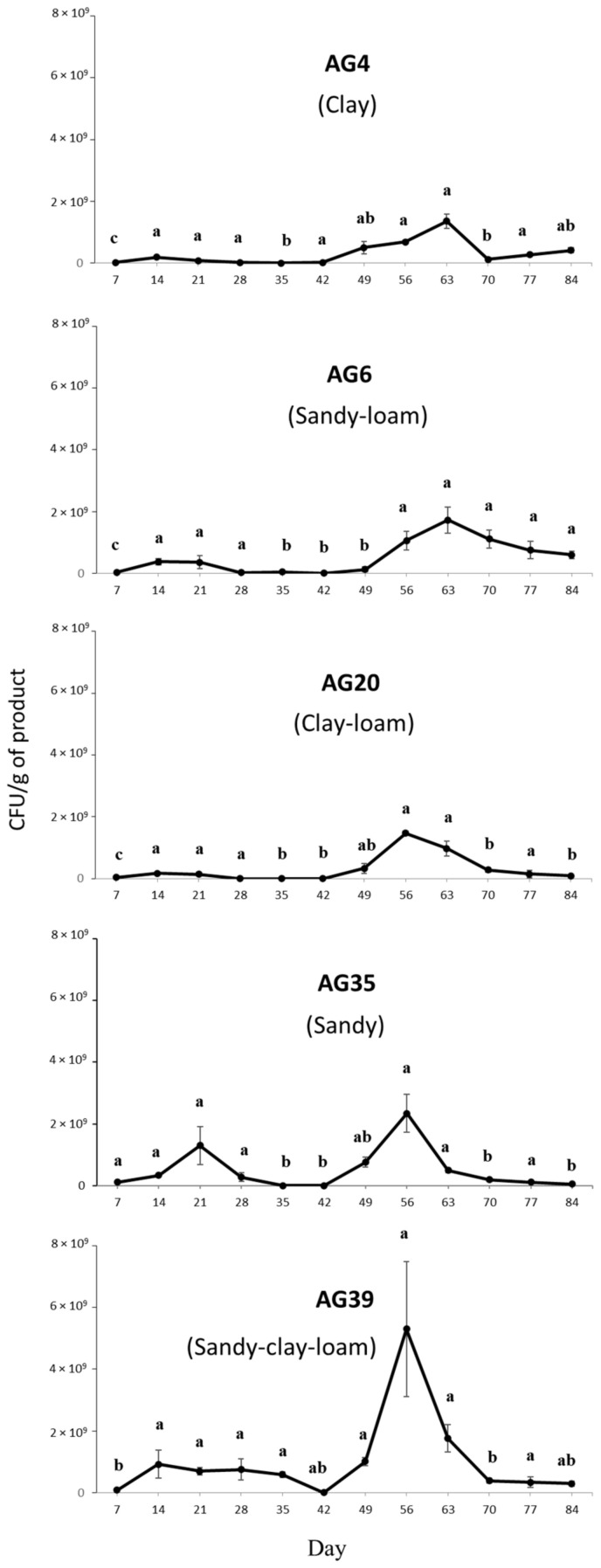
Time course of CFU yield per gram of *M. brunneum* strain EAMa 01/58-Su microsclerotia, according to the type of soil, over a period of 3 months. Mean values (±SE) within the evaluation period (days) followed by different letters are significantly different to one another according to Fisher’s protected Tukey’s HSD test (*p* ≤ 0.05).

**Figure 6 jof-08-00402-f006:**
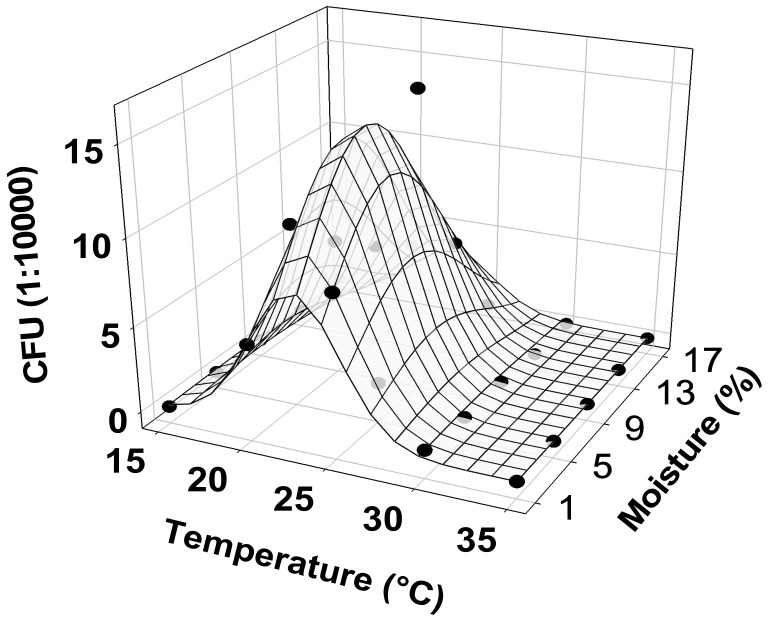
Effects of temperature and moisture on the CFU yield per gram of *M. brunneum* strain EAMa 01/58-Su microsclerotia. The mesh plots represent the predicted values, and the dots represent actual data.

**Table 1 jof-08-00402-t001:** Strains used in the study of microsclerotia production.

Strains *	Fungal Species	Location	Ecosystem of Isolation
EAMa 01/158-Su (CECT 20987)	*M. robertsii*	Utrera (Seville)	Olive orchard
EAMb 09/01-Su (CECT 20784)	*M. brunneum*	Castilblanco de los Arroyos (Seville)	Holm oak dehesa
EAMa 01/58-Su (CECT 20764)	*M. brunneum*	Hinojosa del Duque (Cordoba)	Wheat

* Strains deposited in the Spanish Collection of Culture Types (CECT), with accession numbers included in parentheses.

**Table 2 jof-08-00402-t002:** Geographical location and properties of the soil samples used.

Soil Reference	Geographical Location		Soil Type	Soil Composition			
	Locality	Province		Sand (g/Kg)	Silt (g/Kg)	Clay (g/kg)	Textural Class
AG4	Córdoba	Córdoba	Inceptisol	224	318	458	Clay
AG6	Obejo	Córdoba	Entisol	660	230	110	Sandy loam
AG20	Luque	Córdoba	Inceptisol	405	245	350	Clay loam
AG35	Pozoblanco	Córdoba	Alfisol	860	90	50	Sandy
AG39	Fuente Obejuna	Córdoba	Alfisol	630	110	260	Sandy clay loam

**Table 3 jof-08-00402-t003:** Parameter estimates of the effects of temperature and soil moisture on CFU yield per gram of *M. brunneum* strain EAMa 01/58-Su microsclerotia.

Factor	Parameter	Estimate	Std. Error	Likelihood Ratio χ^2^ (df = 1)	Probability
Intercept	*β* _0_	−35.3	9.1	56.9	<0.0001
Temperature	*β* _1_	3.3	0.8	66.6	<0.0001
Moisture	*β* _2_	0.2	0.1	4.3	0.0389
Temperarture^2^	*β* _4_	−0.1	0	70.9	<0.0001
Moisture^2^	*β* _5_	0	0	6.5	0.011

*Log* (*CFU* 1:10000) = *β*_0_ + *β*_1_*temperature* + *β*_2_*moisture* + *β*_3_*temperature* × *moisture* + *β*_4_*temperature*^2^ + *β*_5_*moisture*^2^.

**Table 4 jof-08-00402-t004:** Effect of UV-B radiation (1200 mWm^−2^) exposure time on CFU yield by *M. brunneum* strain EAMa 01/58-Su microsclerotia.

Treatment (h)	CFU ± SE (×10^7^)
UV-B 0	9.3 ± 5.4 a
UV-B 4	91.0 ± 59.0 a
UV-B 8	6.0 ± 2.0 a
UV-B 24	17.0 ± 5.4 a
UV-B 48	28.0 ± 9.6 a
UV-B 72	41.0 ± 29.0 a

CFU production values by the microsclerotia followed by the same letter indicate that there were no significant differences according to Tukey’s HSD test (*p ≤* 0.05).

## Data Availability

The data that support the findings of this study are available from the corresponding author upon reasonable request.

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
