# Peer review of "Production of Microsclerotia by Metarhizium sp., and Factors Affecting Their Survival, Germination, and Conidial Yield"

_jof, 2022, doi:10.3390/jof8040402_

Round 1

Reviewer 1 Report

The authors tried to characterize microsclerotia (MS) production of two Metarhizium brunneum and one M. robertsii strains. And the effects of storage temperature, incubation time, soil texture, soil temperature and humidity, and ultraviolet radiation (UV-B) on MS germination and infective propagules yield of M. brunneum were determined. Their findings are important for handling and storage of MS and for decision making on MS dosage and timing of application. I provided some comments for the authors to consider as outlined below.

General comments:

  1. It is better to provide the pictures of microsclerotia three tested strains in Figure 1. Because of different diameter of MS having different ability in infective propagules yield, the author should mention whether there is difference diameter of MS, produced by three strains in the results 3.1. Moreover, it is also better to present the effects data of storage temperature, incubation time, soil texture, soil temperature and humidity, and ultraviolet radiation (UV-B) on MS germination and infective propagules yield of the other two strains.
  2. The result showed that EAMa 01/58-Su strain having the highest number at 1x10 7 CFU/g MS. However, after mathematical formula process with Log. Compared with the data of Figure 3, there is lower CFU ability. Why?
  3. There was no significant effect of MS exposure to UV-B 1200mWm-2 on their germination and CFU yield. However, these data was different previous investigation in Nomuraea rileyi (Optimization of culture medium for microsclerotia production by Nomuraea rileyi and analysis of their viability for use as a mycoinsecticide. BioControl 2014, 59:597-605) and in Purpureocillium lilacinum (Liquid culture production of microsclerotia of Purpureocillium lilacinum for use as bionematicide. Nematology 2016, 18(6):719-726). The authors should discuss in Discussion.

Other comments:

  1. P value should be italic.
  2. References should be revised carefully.
  3. Line345: 4℃.
  4. Line 385-390: The summary should rewrite.

Author Response

Dear Reviewer 1

The reviewers’ comments are in italics and underlined, whereas our response is in normal font.

Response to Reviewer 1 Comments

The authors tried to characterize microsclerotia (MS) production of two Metarhizium brunneum and one M. robertsii strains. And the effects of storage temperature, incubation time, soil texture, soil temperature and humidity, and ultraviolet radiation (UV-B) on MS germination and infective propagules yield of M. brunneum were determined. Their findings are important for handling and storage of MS and for decision making on MS dosage and timing of application. I provided some comments for the authors to consider as outlined below.

Dear reviewer 1, Thank you very much for your opinion and valuable review.

General comments:

1. It is better to provide the pictures of microsclerotia three tested strains in Figure 1. Because of different diameter of MS having different ability in infective propagules yield, the author should mention whether there is difference diameter of MS, produced by three strains in the results 3.1.

Done. We provided a new figure 1 with the pictures of MS the three tested strains. In general, the MS of the three strains have similar diameter. We apologize for this detail, but however, our main objective was to determine the capacity of the MS to produce infective propagules by the same unit of quantity (1g of MS). We will seriously take into account this issue for our future studies.

Moreover, it is also better to present the effects data of storage temperature, incubation time, soil texture, soil temperature and humidity, and ultraviolet radiation (UV-B) on MS germination and infective propagules yield of the other two strains.

We agree with the reviewer comment about the importance of including the three strains in all the assays of the study. However, this was challenging for us one year ago. From one hand, some issues of MS production related to the capacity to produce quantities sufficient to cover all assays. For that, we used many agitators at the same time and the process to obtain the MS particles was a bit long and difficult. For that, we decided to select one strain based on their capacity to produce MS and the capacity of these MS to produce infective propagules as shown in figure 2 (A and B), as no difference was shown in the production of MS/l in each of the strains evaluated, but a significant difference was shown in the production of CFU/g of MS.  Also, the selected strain (EAMa01/58-Su) has been demonstrated in many previous studies high efficacy in soil application for the control of soil dwelling stages of some important insect pests at field level. For all that, we decided to continue working only with the EAMa 01/58-Su strain, which is the one that gave us the best results. Anyhow, as we mentioned that we will include these strains in our future studies since they demonstrated to be also promised strains.

2. The result showed that EAMa 01/58-Su strain having the highest number at 1x10 7 CFU/g MS. However, after mathematical formula process with Log. Compared with the data of Figure 3, there is lower CFU ability. Why?

Dear reviewer, you are absolutely right with this observation. There was a misunderstanding occurred writing the M&M of this assay (section 2.3 effect of storage temperature). In this assay we calculated the production of conidia and not CFU. For that, we used another mythological process as clarified in lines 128-131 of the new version of the manuscript (To determine the number of conidia/g MS, germinated MS (with conidia) were scraped from Petri plates into a sterile solution of 0.1% Tween 80, sonicated (Ultrasons HD 3,000,865, J.P. Selecta S.A., Barcelona, Spain) for 5 min and then filtered through severa layers of sterile gauze. Then, serial dilutions were done. Conidia concentration was determined with a haemocytometer, (Malassez chamber, Blau Brand, Wertheim, Germany).  We used this procedure in this assay since it was easier and faster to see and count conidia directly by a haemocytometer compared to the other assays in which we used soil and then, we should count CFU due to the difficulties to see conidia jointly with soil propagules. Then, as you can observe, there has been an error in the graphical representation of figure 3. What we have called CFU would be conidia and therefore this would explain the difference in production that there is for this strain per grams of MS with respect to the production obtained in the figure 2B. Because it would be necessary to explain that a spore gives rise to a CFU, but a CFU can be composed of more than one conidia, this would explain the higher production that appears in figure 3 even if a logarithmic formula is applied. (Breed, R.; Dotterrer, W.D. The number of colonies allowable on satisfactory agar plates. J. Bacteriol. American Society for Microbiology. 1916, 1, 321-331; Goldman, E., Green, L. H. Quantification of microorganisms. In Practical handbook of microbiology, 2nd ed.; Lee, P.S.., Ed.; CRC Press: Nueva York; NY, USA, 2008; p. 18).

Done. Changes in Figure 3 and epigraphs 2.3 and 3.2.

3. There was no significant effect of MS exposure to UV-B 1200mWm-2 on their germination and CFU yield. However, these data were different previous investigation in Nomuraea rileyi (Optimization of culture medium for microsclerotia production by Nomuraea rileyi and analysis of their viability for use as a mycoinsecticide. BioControl 2014, 59:597-605) and in Purpureocillium lilacinum (Liquid culture production of microsclerotia of Purpureocillium lilacinum for use as bionematicide. Nematology 2016, 18(6):719-726). The authors should discuss in Discussion.

Done. Both papers were added to the manuscript discussion and to the reference list. line 468, reference 40 and line 642, reference 56.

Other comments:

  1. P value should be italic. Done.
  2. References should be revised carefully. Done.
  3. Line345: 4℃. Done.
  4. Line 385-390: The summary should rewrite. Done

Reviewer 2 Report

The manuscript to explore the potential of three strains to produce Microsclerotia for further investigation to evaluate the impact of storage temperature, incubation time, UV, soil texture, moisture, and temperature, on the germination and infective propagules is of preliminary study. The manuscript might become an interesting study if the authors

  1. Perform more experimentation to correlate infectivity with virulence by calculating/analyzing enzymatic regulations involved in virulence.
  2. Performing concentration-mortality response bioassays and time-mortality-response bioassays to justify and correlate infectivity; as germination assays performed in this study are not fully depicting the infectivity scope of any pathogen.
  3. What was the criterion to select only these three strains of entomopathogenic fungi?
  4. I noticed errors in the lettering such as Fig 2B, how it is possible that lower CFU values to be used letter (a), and higher with (b)? Please justify otherwise remove typos.
  5. Statistical analysis must be revised and better to reanalyze especially data plotted in Figure 3.
  6. Figure captions are not clearly better to elaborate by providing details about the experimental design used to analyze such data.
  7. Figure 4 and 5: Why only one strain (M. brunneum strain EAMa 01/58-Su) was used in this analysis? The authors should provide a comparative analysis of all the tested strains.
  8. The conclusion is a bit weak. Without morality analysis hard to claim it is an excellent candidate, it seems an exaggeration.
  9. The introduction of the study must be improved.
  10. I noticed 26 % Plagiarism, better to remove Plagiarism by rewording the text.

Author Response

Dear Reviewer 2

The reviewers’ comments are in italics, whereas our response is in normal font.

Response to Reviewer 2 Comments

The manuscript to explore the potential of three strains to produce Microsclerotia for further investigation to evaluate the impact of storage temperature, incubation time, UV, soil texture, moisture, and temperature, on the germination and infective propagules is of preliminary study. The manuscript might become an interesting study if the authors.

1. Perform more experimentation to correlate infectivity with virulence by calculating/analyzing enzymatic regulations involved in virulence.

2. Performing concentration-mortality response bioassays and time-mortality-response bioassays to justify and correlate infectivity; as germination assays performed in this study are not fully depicting the infectivity scope of any pathogen.

Dear reviewer 2, thank you for your revision of our manuscript. We think that the term preliminary should be replaced by basic studies. To conduct more advanced studies, we need this first step of ¨preliminary¨ or basic studies in which we should characterize first, the capacity of our strains to produce MS and then the capacity of these MS to produce infective propagules. We also provided novel data about the performance of the MS on different soil types and a modulated the combination and temperature effect on the MS. Also, the storage temperature effect for one year. We agree of course with you about the need to perform more studies regarding virulence and enzymatic regulations and the effect of the concentration etc… We confirm to you that majority of these studies are now ongoing and will be published in a second paper. Indeed, we are carrying out experiments at field level. Please consider the length of any paper, this should be moderated and not intensive. The assays of this paper last at least one year of experimentation.

3. What was the criterion to select only these three strains of entomopathogenic fungi?

Dear reviewer, as indicated in M&M section 2.1, these strains are selected based on our previous studies in which they demonstrated to be excellent candidates to be used against soil dwelling stages of insect pests. As you may know that the microsclerotia are propagules that fit very good with soil application strategies. Please see our references cited in section 2.1.

4. I noticed errors in the lettering such as Fig 2B, how it is possible that lower CFU values to be used letter (a), and higher with (b)? Please justify otherwise remove typos.

Done, we changed.

5. Statistical analysis must be revised and better to reanalyze especially data plotted in Figure 3.

To analyze the data generated from this experiment we used a linear mixed model for repeated measures (split plot in time). This arrangement occurs when you have an experiment where you collect data from the same experimental unit over a series of dates.

The number of CFUs was log10 (x+1) transformed to meet normality and homogeneity of variance assumptions. The model was estimated using the restricted maximum likelihood (REML) method, and means were compared using Tukey’s test (α=0.05). What this analysis explains is that, throughout the 12 months study period, significant differences in conidia production were found only at the first quarter, with a higher yield at -80°C than at the other temperatures.

6. Figure captions are not clearly better to elaborate by providing details about the experimental design used to analyze such data.

Dear reviewer, we provided the detailed experimental design in each section or experiment and the detailed data analyses in the section 2.8. If we provide such information in each figure caption, we will provide a huge figure captions difficult to adapt to the journal and also difficult to understand by the reader.

7. Figure 4 and 5: Why only one strain (M. brunneum strain EAMa 01/58-Su) was used in this analysis? The authors should provide a comparative analysis of all the tested strains.

We agree with the reviewer comment about the importance of including the three strains in all the assays of the study. However, this was challenging for us one year ago. From one hand, some issues of MS production related to the capacity to produce quantities sufficient to cover all assays. For that, we used many agitators at the same time and the process to obtain the MS particles was a bit long and difficult. For that, we decided to select one strain based on their capacity to produce MS and the capacity of these MS to produce infective propagules as shown in figure 2 (A and B), as no difference was shown in the production of MS/l in each of the strains evaluated, but a significant difference was shown in the production of CFU/g of MS.  Also, the selected strain (EAMa01/58-Su) has been demonstrated in many previous studies high efficacy in soil application for the control of soil dwelling stages of some important insect pests at field level. For all that, we decided to continue working only with the EAMa 01/58-Su strain, which is the one that gave us the best results. Anyhow, as we mentioned that we will include these strains in our future studies since they demonstrated to be also promised strains.

8. The conclusion is a bit weak. Without morality analysis hard to claim it is an excellent candidate, it seems an exaggeration.

Done, conclusion changed.

9. The introduction of the study must be improved.

Dear reviewer, please could you provide more details about this point? We cited in our introduction all (or at least majority) of scientific papers on the field of MS. Also, we highlighted the problem of abiotic factors and their importance for the production, formulation and field application of these propagules.

10. I noticed 26 % Plagiarism, better to remove Plagiarism by rewording the text.

Dear reviewer, we carefully checked this point by using the official tool of the University of Cordoba, the application of Turnitin. As you know, there is a huge difference between similarity and plagiarism. We can send you the official report of Turnitin in which you can observe that the similarity of some repeated words such as the name of the fungus, the name of the strain, a small methodological phrase from our own previous works. NO similarity was detected neither in introduction nor in discussion sections. Please could you please provide the report of similarity?

Reviewer 3 Report

The manuscript entitled "Production of microsclerotia by Metarhizium sp. and factors affecting their survival, germination and conidial yield" is appropriate for the journal. It is an original and relevant contribution to generate knowledge about the expectations of using MS as microbial control agents. In the body of the writing, specific comments are made, which the authors must attend to.

Author Response

Dear reviewer 3

The reviewers’ comments are in italics and underlined, whereas our response is in normal font.

Response to Reviewer 3 Comments

The manuscript entitled "Production of microsclerotia by Metarhizium sp. and factors affecting their survival, germination and conidial yield" is appropriate for the journal. It is an original and relevant contribution to generate knowledge about the expectations of using MS as microbial control agents. In the body of the writing, specific comments are made, which the authors must attend to.

Dear reviewer 3, thank you very much for your comments and your revision. We addressed one by one all your comments and suggestion within the manuscript (new version submitted). From other hand, here we detailed responses of your comments.

Use the multiplication symbol (×), instead of x. Replace throughout the document, where necessary.

Done.

Use the International System of Units, to describe base units, in addition to hour, h

Done, changed along the manuscript.

-Section 2.2. The reviewer 4 said: “A very specific culture medium is its production commercially viable? Some cost-benefit comparison in the discussion section what more research is required”.

This liquid medium is viable as some authors indicate in references: 7,8,9,10,12,13.

What happens is that under laboratory conditions in flasks it is tedious to use. It is best to use large-scale fermenters where it has been shown to be faster to produce and produce less waste than a solid medium. However, due to the low numbers of studies addressing this type of propagules, we cannot provide economic data. We agree that more studies to optimize the liquid medium may be of great interest for commercial production.

-Section 3.3. What is this behavior due to? Not discussed.

  1. We think that this is due to the viability loss of the conidia produced initially (at 7, 8 and 9 days). As you know, the MS germinate and produce conidia gradually on the surface of the MS. These conidia have no contact with the culture medium and will lose viability some days later (at 25°C) if no contact is produced with culture medium or insect host. However, in the discussion section, this assay appears form line 481 onward.
  2. Table 4. Done.

Reviewer 4 Report

Microsclerotia is great biocontrol agent for the control of insect pest, especially for the soil-dwelling stages of geophilic pests. This study evaluates the potential of microsclerotia produced by two Metarhizium brunneumand one M. robertsii under different storage conditions. The best temperature, soil texture and humidity are determined for the microsclerotia storage, production, and germination by serial experiments, which provide beneficial insights for this microsclerotia application.

  1. Microsclerotia exhibits great potential for the control of pest, indicating that the final purpose is applying for pest control. The authors evaluated effects of temperature and soil texture on microsclerotia survival and yield, while there was no evaluation on the control potential of this microsclerotia for any pests.
  2. CFU yield of microsclerotia reaches a peak on 11 days. Why does it decrease from 11 to 14 day?
  3. Figure 4, the legend should be “108
  4. In figure 5, CFU yield of M. brunneum microsclerotia is fluctuating from 42 to 84? More explanations are needed.

Author Response

Dear reviewer 

The reviewers’ comments are in italics and underlined, whereas our response is in normal font.

Response to Reviewer 4 Comments

Microsclerotia is great biocontrol agent for the control of insect pest, especially for the soil-dwelling stages of geophilic pests. This study evaluates the potential of microsclerotia produced by two Metarhizium brunneum and one M. robertsii under different storage conditions. The best temperature, soil texture and humidity are determined for the microsclerotia storage, production, and germination by serial experiments, which provide beneficial insights for this microsclerotia application.

Dear reviewer 4, thank you very much for your revision and your comments.

1. Microsclerotia exhibits great potential for the control of pest, indicating that the final purpose is applying for pest control. The authors evaluated effects of temperature and soil texture on microsclerotia survival and yield, while there was no evaluation on the control potential of this microsclerotia for any pests.

Thanks for the comment. It is really a very important point since the final objective is to use these propagules for the control of insect pests. Yes, we are doing this at both laboratory and field level. We will include all these studies in our next paper.

2. CFU yield of microsclerotia reaches a peak on 11 days. Why does it decrease from 11 to 14 day?

We think that this is due to the viability loss of the conidia produced initially (at 7, 8 and 9 days). As you know, the MS germinate and produce conidia gradually on the surface of the MS. These conidia have no contact with the culture medium and will lose viability some days later (at 25°C) if no contact is produced with culture medium or insect host. However, in the discussion section, this assay appears form line 481 onward.

3. Figure 4, the legend should be “108.

 Done.

4. In figure 5, CFU yield of M. brunneum microsclerotia is fluctuating from 42 to 84? More explanations are needed.

This is a bit like the reason given for comment 3. In this experiment, MS were incubated in soil at field capacity of humidity. MS were germinated gradually and produced conidia overtime. That’s why we begun to detect conidia from the first week and this was increased up to 8 weeks. These conidia in will lose viability and the detected concentration will be decreased overtime. The decrease of inoculum detection in the soil overtime is cited by our previous works from the first month after application of conidia (Yousef et al., 2017, 2018). However, these results have been further discussed in lines 626 onward.

Round 2

Reviewer 1 Report

Thank you for response.

Author Response

Thank you very much

Reviewer 2 Report

I am sorry to write that the authors did not follow my suggestions. They only incorporate a few of my suggestions. But, the main concern was about the analysis, more experimentation needs to be addressed by reanalyzing and performing experiments. Without taking into consideration my previous comments, the article seems to me a preliminary study that does not merit to be published in such esteemed journal "Journal of Fungi". Lastly, I have suggested lettering but I noticed that they randomly allocated lettering without applying Tukey HSD test. I have serious doubts, therefore, I am not in a position to accept it until the authors seriously revise their manuscript.
